# Unconventional magnetism mediated by spin-phonon-photon coupling

Petros Andreas Pantazopoulos [1] ✉, Johannes Feist [1] ✉,
Francisco J. García-Vidal [1] ✉ & Akashdeep Kamra [1] ✉

Magnetic order typically emerges due to the short-range exchange interaction between the constituent electronic spins. Recent discoveries have found a crucial role for spin-phonon coupling in various phenomena from optical ultrafast magnetization switching to dynamical control of the magnetic state. Here, we demonstrate theoretically the emergence of a biquadratic long-range interaction between spins mediated by their coupling to phonons hybridized with vacuum photons into polaritons. The resulting ordered state enabled by the exchange of virtual polaritons between spins is reminiscent of superconductivity mediated by the exchange of virtual phonons. The biquadratic nature of the spin-spin interaction promotes ordering without favoring ferro- or antiferromagnetism. It further makes the phase transition to magnetic order a first-order transition, unlike in conventional magnets. Consequently, a large magnetization develops abruptly on lowering the temperature which could enable magnetic memories admitting ultralow-power thermally-assisted writing while maintaining a high data stability. The role of photons in the phenomenon further enables an in-situ static control over the magnetism. These unique features make our predicted spin-spin interaction and magnetism highly unconventional paving the way for novel scientific and technological opportunities.

Magnets host a broad range of intriguing ground states from spin liquids[1] to topological textures[2], such as skyrmions. These play a central role in the various correlated states of electronic matter and subfields of physics, like spintronics[3] and unconventional superconductivity[4]. Magnets have also had a tremendous impact on contemporary computing technology via magnetic random access memories[5] and read heads based on the magnetoresistance effects[6]. Different kinds of order, such as ferromagnetic and antiferromagnetic, emerge primarily due to the short-range exchange interaction between neighboring spins, which get aligned parallel or antiparallel to each other.

While the exchange is typically the strongest interaction in magnets, spin-lattice or spin–phonon coupling has been found to underlie the transfer of spin angular momentum between the magnetic and lattice degrees of freedom[7]. Phenomena such as ultrafast optical switching of magnetic moments[8–13] and long-range transport of spin via phonons[14–20] rely fundamentally on the spin–phonon interaction. Another of its key consequences is mediating a linear[14–21] or nonlinear[21–23] coupling between phonons and magnons - the spin excitations of ordered magnets. In these considerations, a pre-existing exchange interaction underlies the magnetically ordered ground state, while the spin–phonon interaction enables mutual coupling and control between the excitations. Spin–phonon coupling has also been exploited to dynamically control the magnetic state via optically driving certain phonon modes[13,24–26], similar to recent schemes employing light as a strong drive to achieve control over magnetic or even nonmagnetic states of matter, such as a superconductor[27].

[1]Departamento de Física Teórica de la Materia Condensada and Condensed Matter Physics Center (IFIMAC), Universidad Autónoma de Madrid, Madrid E-28049, Spain. ✉e-mail: petros.pantazopoulos@uam.es; johannes.feist@uam.es; fj.garcia@uam.es; akashdeep.kamra@uam.es

At the same time, modifying existing ordered states or phase transitions by tuning the equilibrium electromagnetic environment is currently a highly desired and pursued goal[28,29]. Along these lines, the linear-in-spin coupling with light has been predicted to mediate quadratic spin–spin interactions mimicking antiferromagnetic exchange and stabilizing spin liquid states[30,31]. A paradigmatic work[32] considered a similar linear-in-pseudospin coupling with phonons to demonstrate a quadratic pseudospin-pseudospin interaction examining its effect on the system's dynamical properties. Such a potential linear-in-spin coupling with phonon displacement is forbidden by time-reversal symmetry[14].

Although spin–phonon coupling has recently been established to play an important part in a large number of nonequilibrium spin phenomena, its potential role in determining the fundamental interaction and ground state of a magnet has not been explored. Here, we theoretically demonstrate the emergence of an unconventional long-range, algebraically decaying interaction between localized spins due to the exchange of virtual phonons coupled to the vacuum photon modes. The basic phenomenon is similar to how electron-electron attraction emerges in a metal from the exchange of virtual phonons, leading to superconductivity. It is also reminiscent of van der Waals interactions emerging from virtual charge density fluctuations[33]. On account of the spin–phonon coupling being quadratic-in-spin due to time-reversal invariance, the emergent spin–spin interaction is found to be biquadratic in the spin components, in contrast with the quadratic nature of the conventional exchange interaction. Thus, depending on the sign of the interaction, it enforces order or disorder in the magnet without explicitly favoring parallel or antiparallel configuration. The possibility of promoting disorder can help stabilize spin liquid states[1,34] at higher temperatures. Considering the emergence of ferromagnetism i.e., parallel alignment of all spins, the system is found to manifest a first-order phase transition to a large magnetization just

below the critical temperature $T_c$. Thus, this unconventional magnet enables a promising possibility for memories which would admit ultralow-power thermally-assisted writing[35,36] of a bit by raising the temperature slightly above $T_c$ and cooling in the presence of a weak applied magnetic field, thereby obtaining a large magnetization along a desired direction. Such a process becomes ineffective with conventional magnets that manifest a second-order phase transition because cooling slightly below the critical temperature yields a small magnetization. Since the latter determines the energy barrier between the two equal-energy bit states, the small magnetization results in unstable and unreliable data storage.

## Emergent spin–spin interaction

We consider ferromagnetic nanoparticles, each one bearing a large spin with $S \gg 1$ due to its magnetically ordered state and an infrared (IR)-active phonon mode with zero wavenumber and THz range frequency confined to the nanoparticle (see Fig. 1). While the spin–phonon interaction couples these two subsystems, the nanoparticles remain independent of each other at this level. Interaction between the different nanoparticles is provided by the electromagnetic photon modes in the system which are delocalized over the entire space and which couple to the IR-active phonon in each of the nanoparticles. The total Hamiltonian for the system thus becomes

$$H = H_\mathrm{S} + H_\mathrm{P} + H_\mathrm{EM} + H_\mathrm{S\text{-}P} + H_\mathrm{EM\text{-}P}, \qquad (1)$$

where $H_\mathrm{S}$ describes each of the independent ferromagnetic nanoparticle spins and may include contributions from Zeeman coupling or any local magnetic anisotropies. $H_\mathrm{P}$ captures the phonon along each Cartesian direction on each of the nanoparticles. $H_\mathrm{EM}$ accounts for the electromagnetic modes of the environment.

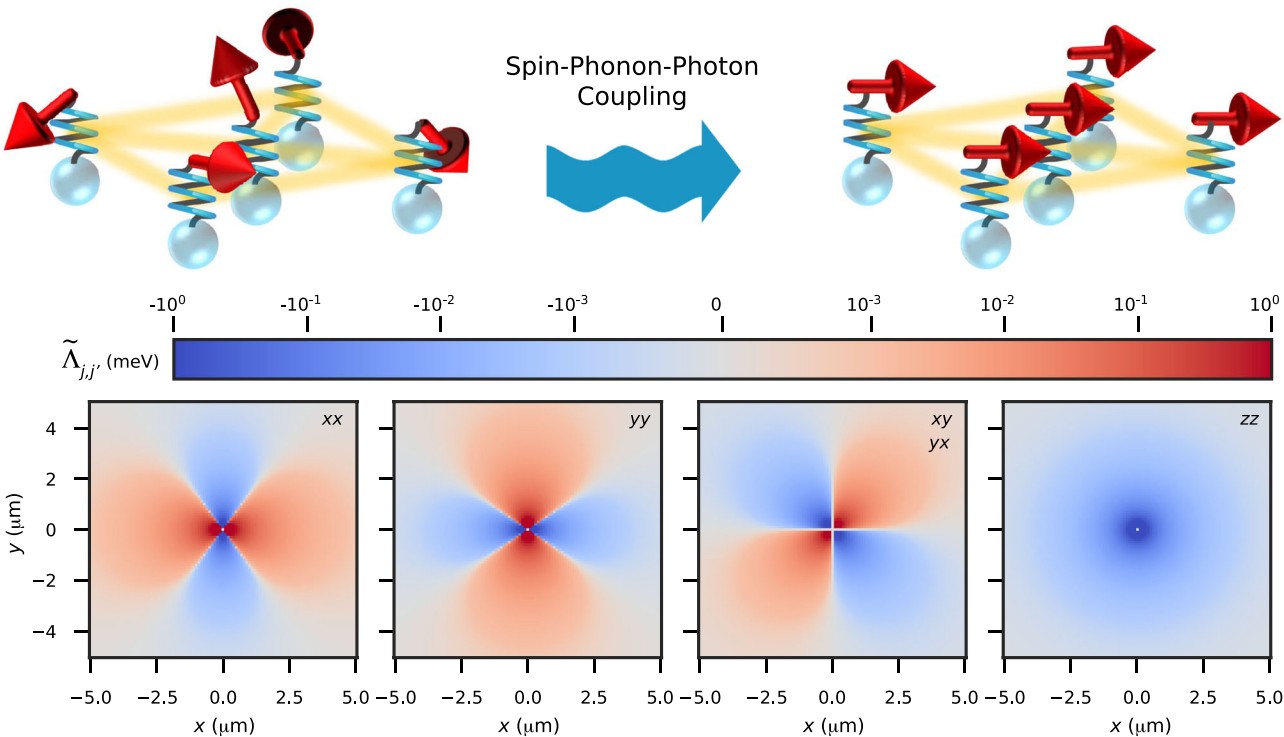

**Fig. 1 | Emergent spin–spin interaction and ordering mediated by spin-phonon-photon coupling.** Top panel: Schematic depiction of the system. Localized spins, illustrated as red arrows, are coupled to local phonons, shown as springs. When we consider the phonons to be coupled with global photons (yellow shading) forming polaritonic modes, an exchange of virtual polaritons between the spins causes an effective spin–spin interaction resulting in an in-plane ordering of the spins. Bottom panel: Non-vanishing components of the spin–spin coupling tensor $\tilde{\Lambda}_{j,j'}$ between the central nanoparticle located at $\mathbf{r}_j$ and the one situated at $\mathbf{r}_{j'}$ in a square array of spheres in the $x$-$y$ plane.

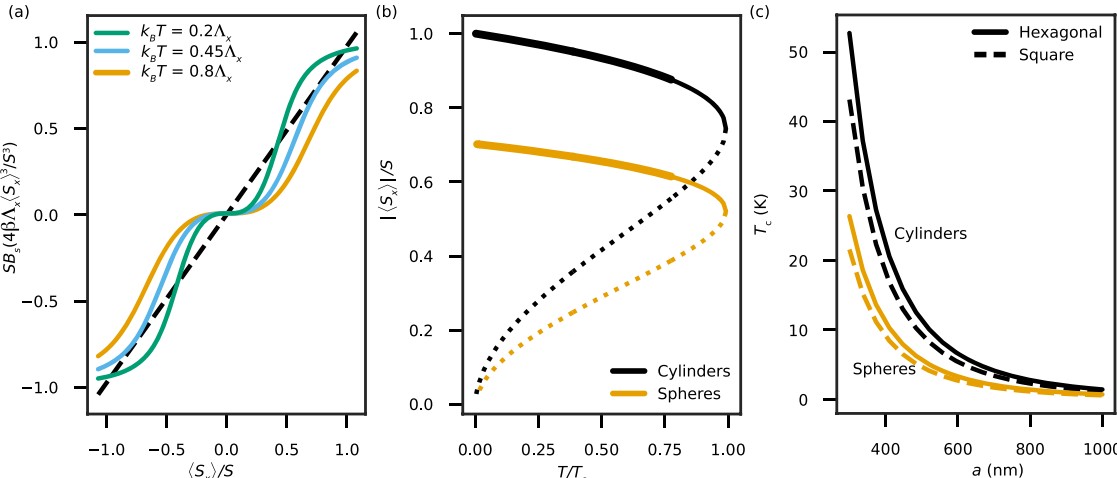

**Fig. 2 | Temperature dependence of the unconventional ferromagnetic state.**
**a** Graphical solution of the self-consistency equation for the mean value of spin $x$ component, $\langle S_x \rangle / S = S B_S \left( 4\beta \Lambda_x \langle S_x \rangle^3 / S^3 \right)$, for various coupling strengths and $S \gg 1$. A solution is obtained when the solid line crosses the dashed line $y = \langle S_x \rangle$. $k_B$ and $T$ are the Boltzmann constant and temperature, respectively. **b** Temperature dependence of the absolute mean value of the $x$ component of the spin for an array of cylindrical (black line) and spherical (orange line) nanoparticles, with solid (dotted) lines indicating locally stable (unstable) solutions. The thick solid lines indicate that the solution furthermore corresponds to the global minimum of the free energy. **c** Dependence of the critical temperature $T_c$ on the lattice constant $a$ of the square (solid line) and hexagonal (dashed line) array of spherical (orange line) or cylindrical (black line) nanoparticles.

The spin–phonon coupling is obtained as

$$H_{\text{S-P}} = \sum_j \sum_{k=x,y,z} b_k \frac{S_{j;k}^2}{S^2} \left( \beta_{jk}^\dagger + \beta_{jk} \right), \tag{2}$$

where $j$ runs over all the nanoparticles, $S_{j;k}$ is the $k$th Cartesian component of the nanoparticle $j$'s spin, $\beta_{jk}$ is the annihilation operator of nanoparticle $j$'s phonon mode polarized along the $k$th Cartesian component, and $b_k$ parametrizes the spin–phonon interaction strength. The form of this coupling has been derived within a simple model in Supplementary Note 1. It is quadratic in spin components due to time-reversal symmetry, while the linear coupling to an IR-active phonon necessitates a noncentrosymmetric ferromagnet[37,38], such as $Cu_2OSeO_3$. Equation (2) is obtained by quantizing the classical magnetoelastic coupling Hamiltonian[14,16], appropriately generalized to optical phonons (see Supplementary Note 1), in terms of the phonon ladder operators. The effective spin–phonon coupling $b_k$ can be expressed in terms of the material parameters (see Supplementary Note 1). Much of the unconventional nature of the ensuing spin–spin interaction and magnetism is a direct consequence of the spin–phonon coupling [see Eq. (2)] being invariant under time-reversal, such that it remains the same when **S** is replaced by −**S**. As a result, our main results discussed below are independent of the model details.

The nanoparticle phonon modes couple to the electromagnetic photon modes via the dipole-electric field interaction

$$H_{\text{EM-P}} = \sum_{n,j} \sum_{k=x,y,z} \mathbf{d}_{jk} \cdot \mathbf{E}_n(\mathbf{r}_j) \alpha_n^\dagger \beta_{jk} + \text{H.c.}, \tag{3}$$

where $n$ runs over all the electromagnetic modes, $\alpha_n$ is the annihilation operator for photon mode $n$ and $\mathbf{E}_n(\mathbf{r})$ is its electric field. The position vector of nanoparticle $j$ is $\mathbf{r}_j$, with $d_j$ the dipole moment of its phonon modes, and $\mathbf{d}_{jk} = d_j \hat{\mathbf{k}}$. We have further employed the rotating wave and long-wavelength (see Supplementary Note 7) approximations in considering the coupling between the photons and the zero-wavenumber IR-active phonons.

Diagonalization of the phonon plus electromagnetic Hamiltonian yields polaritonic modes (see Supplementary Note 2) that are delocalized over the whole system. Consequently, the nanoparticle spins interact with the common polaritons in the environment. We obtain the following effective spin Hamiltonian that describes the exchange of virtual bosons by integrating out the polariton modes employing the path integral framework and evaluating the canonical partition function (see Methods, Supplementary Notes 2, 3)

$$H_{\text{eff}} = H_{\text{S}} - \frac{1}{S^4} \sum_{jj'} \mathbf{S}_j^2 \cdot \tilde{\boldsymbol{\Lambda}}_{jj'} \cdot \mathbf{S}_{j'}^2, \tag{4}$$

where $\mathbf{S}^2 \equiv (S_x^2, S_y^2, S_z^2)$ denotes a vector made by the spin component squares. $\tilde{\boldsymbol{\Lambda}}$ is a $3 \times 3$ tensor describing the coupling in units of energy between the different Cartesian components and depends on the composition of the polaritonic modes and their coupling to the spins. Treating the full continuum of electromagnetic modes (see Supplementary Note 4), we obtain

$$\tilde{\Lambda}_{jj'} = \frac{b^2 d_j d_{j'}}{2\epsilon_0 c^2 \hbar^2 \Omega^2} \text{Re} \left[ \omega^2 \tilde{\mathbf{G}}(\mathbf{r}_j, \mathbf{r}_{j'}, \omega) \right]_{\omega=0}, \tag{5}$$

where $\Omega$ is the phonon frequency (assumed to be the same for all nanoparticles), $\bar{\mathbf{G}}$ is the dyadic Green's function of the electromagnetic field, $\epsilon_0$ is the vacuum electric permittivity, $c$ is the vacuum speed of light, and the spin–phonon coupling is assumed to be isotropic, i.e., $b_k = b$ for all $k$. This elegant formula is not restricted to a particular geometry, and thus it allows for studying complex structures while enabling the design of optimized systems. The strength of the spin–spin coupling depends on optical, phononic, and magnetoelastic material parameters as well as on the electrostatic ($\omega = 0$) response of the electromagnetic environment. Notably, it already acts in free space, and does not rely on the presence of a cavity of any kind, nor on achieving strong coupling between light and matter resonances[39], nor on other resonant effects.

Equations (4) and (5) constitute one of our main results and demonstrate an emergent interaction between the spins. For positive (negative) $\tilde{\Lambda}$ components, energy is minimized by having a large

(a)

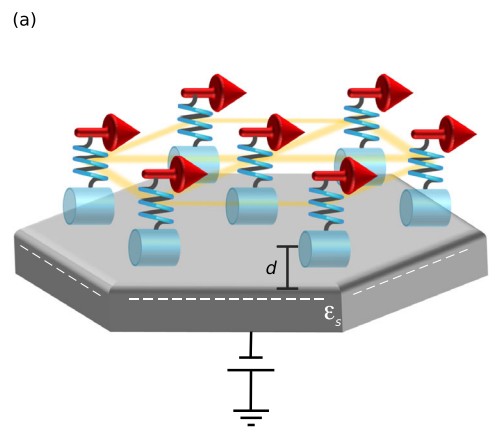

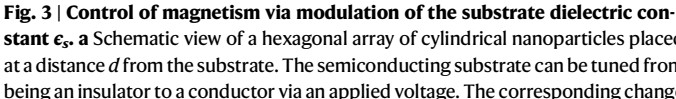

(b)

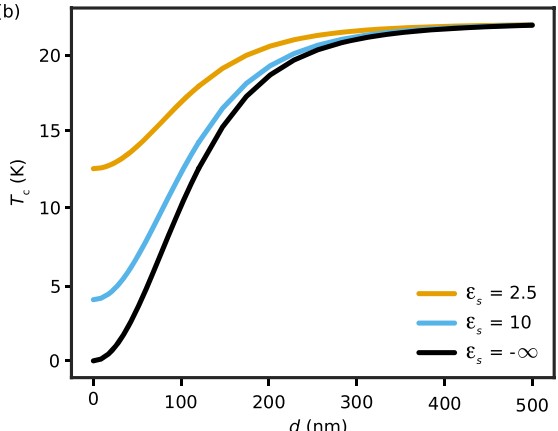

**Fig. 3 | Control of magnetism via modulation of the substrate dielectric constant $\epsilon_s$. a** Schematic view of a hexagonal array of cylindrical nanoparticles placed at a distance $d$ from the substrate. The semiconducting substrate can be tuned from being an insulator to a conductor via an applied voltage. The corresponding change

in its static dielectric constant alters the polariton modes that mediate the spin–spin interaction. **b** Dependence of the critical temperature $T_c$ of the unconventional ferromagnetic state on $d$ when the substrate is an electrical insulator ($\epsilon_s = 2.5$; orange line, $\epsilon_s = 10$; blue line) or a conductor (black line).

(zero) value of the corresponding spin component squares. Thus, for positive values of the components of $\tilde{\Lambda}$, the emergent interaction encourages ordering of the spins without explicitly preferring ferromagnetic or antiferromagnetic configuration. This symmetry and degeneracy between the two kinds of ordering may be lifted by additional interactions not explicitly considered here. Conversely, a negative value of $\tilde{\Lambda}$ components promotes disorder. This may reinforce effects such as geometrical frustration in spin liquids[1], leading to their higher stability.

Considering a two-dimensional square array of spherical nanoparticles and employing known material parameters (see Supplementary Note 5), we show the various non-vanishing components of $\tilde{\Lambda}$ as a function of the distance between the spins in Fig. 1. A strongly anisotropic nature of the emergent interaction can be seen. While the in-plane ($x$-$y$) components of $\tilde{\Lambda}$ are positive on average, thereby supporting an ordered state, the out-of-plane ($z$) component remains negative for all spins. Thus, the emergent spin–spin interaction encourages the spins to remain in the plane, giving rise to two-dimensional magnetism.

## Mean-field theory of unconventional ferromagnetism

We now examine the emergence of ferromagnetic order due to the spin–spin interaction derived above. To this end, we continue to assume a two-dimensional organization of the nanoparticles in either a square or a hexagonal pattern. Further, we consider spherical or cylindrical nanoparticles. In the former, phonons polarized along any spatial direction interact equally with the electric field. In contrast, for cylindrical particles, the phonon mode polarized along the cylinder axis will couple most strongly to the electric field, so that the interaction can be approximated as being due to a single dipole component.

Assuming all spins to be in the same state due to translational invariance and that no external magnetic field is applied, the spin–spin interaction [see Eq. (4)] results in the following mean-field Hamiltonian (see Supplementary Note 6)

$$H_{\mathrm{MF}} = -\frac{1}{S}\sum_j \mathbf{h}_j \cdot \mathbf{S}_j, \tag{6}$$

with $\mathbf{h}_j = 4(\Lambda_{j;x}\langle S_x\rangle^3 \hat{\mathbf{x}} + \Lambda_{j;y}\langle S_y\rangle^3 \hat{\mathbf{y}} + \Lambda_{j;z}\langle S_z\rangle^3 \hat{\mathbf{z}})/S^3$ the effective magnetic field. Here, $\langle\cdot\rangle$ denotes the expectation value and $\Lambda_{j;x} \equiv \sum_{j'} \tilde{\Lambda}_{j,j'}^{xx}$, and so on. Thus, only the diagonal components of the $\tilde{\Lambda}_{ij}$ tensor

contribute to the net magnetic order in the whole ensemble. We obtain the self-consistency equation for determining the $x$ component of each spin (see Supplementary Note 6)

$$\langle S_x\rangle = \frac{h_x}{h} SB_S(\beta h) \tag{7}$$

and so on for the $y$ and $z$ components where $B_S(x)$ is the Brillouin function and $\beta = 1/(k_B T)$ with $k_B$ the Boltzmann constant and $T$ the temperature. Here, the $j$-dependence of $h_j$ has been dropped assuming translational invariance.

Figure 2a qualitatively shows the graphical solution of the self-consistency Eq. (7) assuming non-vanishing coupling only along the $x$ axis for $S \gg 1$, i.e., $\langle S_x\rangle = SB_S(4\beta\Lambda_x\langle S_x\rangle^3/S^3)$. In contrast with the case of conventional ferromagnetism which admits a unique stable solution to the self-consistency equation, here we find two solutions. This is a direct result of the mean-field $\mathbf{h}$ components scaling as the third power of the corresponding spin component expectation value, instead of the first power as is the case for conventional magnetism.

The absolute value of the two solutions as a function of temperature is displayed in Fig. 2b. We find that the spin expectation value develops a finite and large value abruptly as the temperature is lowered, which corresponds to a first-order phase transition. In contrast, conventional magnetism corresponds to a second-order phase transition in which the magnetization increases gradually as the temperature is lowered below the Curie temperature. The solution associated with a high expectation value is stable, and the other is unstable, because they correspond to a minimum and a maximum in the Helmholtz free energy, respectively. When the temperature decreases further, the stable solution replaces the trivial solution, $\langle S_x\rangle = 0$, as the energetically favorable state since it becomes the global minimum of the free energy. Furthermore, due to the negative value of $\Lambda_{jz}$, the expectation value of the spin $z$ component vanishes, and the spins are oriented in the $x$-$y$ plane.

Figure 2c shows the critical temperature of the ferromagnetic state as a function of the lattice constant for the two kinds of nanoparticle arrays considered, in air, thereby providing guidance for achieving a desired critical temperature by choosing the right arrangement of the nanoparticles. The critical temperature is affected by both the density and the configuration of the lattice. As can also be seen in Fig. 2b, the shape of the nanoparticles modifies the expectation value of the spin $x$ components and the critical temperature, and can

thus be used as an additional degree of freedom to obtain the desired result. The reason is the geometrical factor appearing in the self-consistency equation and specifically in $h$. For the case of cylinders, the effective magnetic field $h$ is proportional to $\Lambda_x$ while for the case of spheres, it is proportional to $\sqrt{2}\Lambda_x$ due to equal contributions from $x$ and $y$ components, yielding a lower critical temperature.

## Static control of magnetism

There has been a tremendous interest in and technological need for controlling magnetic ground states via external knobs. In conventional magnets relying on exchange interaction between the electronic spins, this proves to be a daunting task since it requires controlling the internal states of and interactions between the constituent electrons. Nevertheless, transient control over these has been obtained at ultra-fast timescales by creating a strong nonequilibrium in the participating electrons[13,24–26,40–44]. Since the unconventional magnetism under consideration relies, in part, on the photon modes, it opens the possibility to control the spin interactions and the consequent magnetic order by modifying these modes, which are easily accessible to the outside world. A recent experiment[45] has already observed an ex-situ change in ferromagnetism via certain resonant effects, distinct from our considerations here. We now demonstrate a strong in situ tunability of the ferromagnet critical temperature via an external knob, such as a gate voltage, that alters the electromagnetic environment of the system.

To this end, we consider that the nanoparticles are dispersed in an electrically insulating medium with static permittivity $\epsilon_m$ (with free space corresponding to $\epsilon_m = 1$). Further, the nanoparticles are now deposited on a substrate and placed at a distance $d$ from it [see Fig. 3a]. The photon modes supported by the system are now modified via the electric field boundary condition imposed by the substrate (see Methods). This directly affects the nature of the polaritons mediating the spin–spin interaction and thus the ferromagnetic state. Figure 3b shows the corresponding critical temperature as a function of nanoparticle-substrate separation for insulating and perfectly conducting substrates, where a critical temperature decrease is obtained as the nanoparticles approach the substrate. This is understood by considering that the spin–spin coupling depends on Green's function at zero frequency, which can be decomposed into two contributions: the free space and the scattered. The latter accounts for the effects of environmental engineering. In the case we study here (see Methods), the scattered contribution due to the substrate partially cancels the free-space one, thus decreasing the coupling and, consequently, the strength of the effective mean field, which determines the critical temperature (see Eq. 7). Interestingly, when the nanoparticles approach the conducting substrate, the critical temperature is reduced dramatically and even vanishes in the limit $d \to 0$. Intuitively, the underlying mechanism can be understood as follows. The image dipoles of the particles in the substrate create a field which tends to cancel out the free-space contribution. Since it is possible to tune, for example, a silicon substrate from being effectively an insulator to a conductor using a gate voltage[46], the critical temperature of the ferromagnetic state may be controlled, in situ and in equilibrium, via the applied voltage [see Fig. 3a]. Furthermore, by choosing a specific operating temperature, the gate voltage enables a complete turning on and off of the magnetic order. Besides its potential application in memories, this functionality could enable computing architectures based on magnets or spin waves.

## Discussion

In evaluating the physical observables here, we have employed materials parameters typical of magnetic materials, such as iron garnets and ferrites, as detailed in Supplementary Note 5. We note that in order to evaluate the coupling strength of the spin-phonon-photon coupling described here, a candidate material would need to be characterized in terms of both its optical phonons and magnetoelastic properties. This set of parameters is not available for most materials, presumably in part because this specific combination had not been previously identified as playing an important role. Therefore, the present work suggests that an interesting direction for future research would be to look for materials that maximize the effects described here. For instance, recent experiments demonstrate a significantly enhanced magnon-phonon interaction[47], indicating that our choice of parameters is likely conservative, and better materials would be available.

While we have considered configurations with periodically arranged nanoparticles for concreteness, the qualitative results regarding unconventional ferromagnetism remain the same for arbitrary and disordered two-dimensional arrangements. Hence, it is not crucial for experimental realizations to achieve a perfect pattern. At the same time, our proposal is valid and offers a new dimension to artificial spin liquids[34] by choosing magnets with optimal optical phonon modes and lithographically fabricating the desired configuration. Furthermore, our proposed unconventional magnets based on spin–phonon coupling may enable a more effective and faster means of implementing cooling in nanostructures based on adiabatic demagnetization, especially considering the unconventional first-order nature of the transition.

In conclusion, we theoretically demonstrate an emergent biquadratic spin–spin interaction mediated by spin-phonon-photon coupling. The role of photons in this phenomenon lends a strong anisotropy to the spin–spin interaction making the magnetism two-dimensional as well as enabling static in situ control over the critical temperature of the magnetically ordered state. Since our investigated unconventional spin interaction emerges from an exchange of virtual bosons, much like superconductivity, our work opens new avenues for exploring similarly unconventional spin interactions and magnetism mediated by different bosonic modes available in solid-state systems.

## Methods

### Obtaining polariton modes

The $H_P$, $H_{EM}$, and $H_{EM-P}$ terms constitute a Hamiltonian which can be represented as

$$H_{PP} = \begin{pmatrix} \boldsymbol{\beta}^\dagger & \boldsymbol{\alpha}^\dagger \end{pmatrix} \mathbf{H}_{PP} \begin{pmatrix} \boldsymbol{\beta} \\ \boldsymbol{\alpha} \end{pmatrix} \qquad \text{with } \mathbf{H}_{PP} = \begin{pmatrix} \boldsymbol{\Omega} & \mathbf{g} \\ \mathbf{g}^\dagger & \boldsymbol{\omega} \end{pmatrix}, \qquad (8)$$

where $\boldsymbol{\beta}$ and $\boldsymbol{\alpha}$ are vectors containing the phonon and photon annihilation operators, $\boldsymbol{\Omega}$ and $\boldsymbol{\omega}$ are diagonal matrices describing the energy of the phononic and photonic modes, and $\mathbf{g}$ describes the coupling between the phonons and photons. The eigenvalues of $\mathbf{H}_{PP}$ correspond to the energy of the polaritonic modes, while the eigenvectors contain the coefficients relating the polaritonic operators with the phononic and photonic ones (see Supplementary Note 2).

### Integrating out the polaritons

Equation (2) can be expressed in terms of the polaritons since the phononic operators are related to the polaritonic ones, and thus, the total Hamiltonian reduces to $H_S$, the polaritonic modes, and their interaction with spins. The partition function of the system is calculated on the basis of the spin and polaritonic states. The effective spin–spin interaction is obtained by calculating the contribution due to the polaritonic modes employing the path integral framework[48,49] (see Supplementary Note 3).

The coherent state of each polariton is considered. Since the polaritons do not interact with each other, their contribution to the partition function is a product of Gaussian integrals (one for each mode), which have analytical solutions. For each polaritonic mode, the $j$th spin interacts with the $j'$th spin with strength equal to the product of their couplings with the polariton divided by the polariton's energy. By summing over all polaritonic modes, the spin–spin coupling can then be related to the inverse of $\mathbf{H}_{PP}$. For a continuum of modes, the

latter can be expressed in terms of the dyadic Green's function of the electromagnetic field (see Supplementary Note 4).

## Accounting for a substrate

The dyadic Green's function of a multilayered structure can be efficiently calculated[50]. For a two-layered structure, where the layers are characterized by static electric permittivities $\epsilon_m$ and $\epsilon_s$ and magnetic permeability equal to unity, the coupling reads

$$\tilde{\Lambda}_{j,j'}(\mathbf{r}_j,\mathbf{r}_{j'}) = \frac{d_j d_{j'}}{8\pi\epsilon_0} \frac{b^2}{\hbar^2\Omega^2}\left[\tilde{\mathbf{l}}_0(\mathbf{r}_j-\mathbf{r}_{j'})+\tilde{\mathbf{l}}_{sc}(|\mathbf{r}_{j;\parallel}-\mathbf{r}_{j';\parallel}|,\theta,z_j+z_{j'})\right],$$

$$\tilde{\mathbf{l}}_0(\mathbf{r})=\frac{1}{r^3\epsilon_m}(3\hat{\mathbf{r}}\otimes\hat{\mathbf{r}}-\mathbf{1}),$$

$$\tilde{\mathbf{l}}_{sc}(R,\theta,z)=-\frac{\epsilon_m-\epsilon_s}{\epsilon_m+\epsilon_s}\frac{2z^2-R^2}{2(z^2+R^2)^{5/2}}\begin{pmatrix}1&0&0\\0&1&0\\0&0&2\end{pmatrix}$$
$$+\frac{\epsilon_m-\epsilon_s}{\epsilon_m+\epsilon_s}\frac{3R^2}{2(z^2+R^2)^{5/2}}\begin{pmatrix}\cos(2\theta)&\sin(2\theta)&-2z\cos\theta/R\\\sin(2\theta)&-\cos(2\theta)&-2z\sin\theta/R\\2z\cos\theta/R&2z\sin\theta/R&0\end{pmatrix},$$

with $\mathbf{r}_j,\mathbf{r}_{j'}$ corresponding to nanoparticles in the $\epsilon_m$ layer.

## Data availability

All the information required for reproducing these results has been provided in the main text and supplemental information. The numerical data generated for plotting the figures is available on reasonable request.

## Code availability

The numerical routines employed for generating the data plotted in the figures are available on reasonable request.

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

## Acknowledgements

We thank Diego Fernandez de la Pradilla and Mathias Weiler for helpful discussions. We acknowledge financial support by the Spanish Ministry for Science and Innovation-Agencia Estatal de Investigacion (AEI) through Grants RTI2018-099737-B-I00 (P.A.P., J.F., and F.J.G.-V.), PID2021-125894NB-I00 (J.F. and F.J.G.-V.), PCI2018-093145374 (through the QuantERA program of the European Commission) (P.A.P., J.F., and F.J.G.-V.), CEX2018-000805-M (through the Maria de Maeztu program for Units of Excellence in R&D) (P.A.P., J.F., F.J.G.-V., and A.K.) and RYC2021-031063-I (financed by MCIN/AEI/10.13039/501100011033 and the European Union NextGenerationEU/PRTR) (A.K.), by the European Research Council through Grant No. ERC-2016-StG-714870 (P.A.P. and J.F.), and by the European Union's Horizon Europe Research and Innovation Program through agreement 101070700 (MIRAQLS, J.F. and F.J.G.-V.). F.J.G.-V. acknowledges financial support from the Comunidad de Madrid and the Spanish State through the Recovery, Transformation and Resilience Plan ["MATERIALES DISRUPTIVOS BIDIMENSIONALES (2D)" (MAD2D-CM)-UAM7], and the European Union through the Next Generation EU funds.

## Author contributions

All authors contributed to the calculations, discussions, and writing of the article.

## Competing interests

The authors declare no competing interests.
