## [Peer Review File · Nature Communications]

Reviewers' Comments:

Reviewer #1:

Remarks to the Author:

The authors theoretically studied a so-called "spin-phonon-photon coupling" induced magnetic state in the nanoparticle system. Spin-phonon interaction couples the local spin of the nanoparticle to the vibrational mode, while an electromagnetic field collectively excites all the nanoparticles across the system, leading to an effective ferromagnetic coupling. The topic is interesting, however, I have a few questions upon reading the article.

First, the definition of "spin" in the paper is confusing to me. The spin operator S in the article is defined in the nanoparticle level, i.e. the net spin of all individual magnetic atoms within the nanoparticle. However, the "phonon" is defined in the atomic level when they derive the spin-phonon coupling Hamiltonian in the SI. Thus, the inconsistency between them causes confusion and the validation of H_{s-p} is questionable.

On the other hand, if the "phonon" is about the vibration between neighboring nanoparticles, this is also problematic. Two issues are raised: a) Since their bonding are generally weak, the lattice translational symmetry across the system is easily broken, leading to the failure of a definition of "phonon"; b) Even if the nanoparticles are bound to a 2D periodic lattice array through some external field, e.g. a substrate, they cannot interact with the incident photon through the dipole interaction unless they are charged! In other words, this model does not work for charge-neutral particles. If the nanoparticles carry positive/negative charges, the Coulomb interaction term should be included in the model. Since the Coulomb interaction is generally larger than the long-range magnetic interaction by some orders of magnitude, I doubt whether the current conclusions still hold.

Furthermore, in the lower panel of Fig.1, it shows that for a two-spin system, the xx (yy) component of $\Lambda_{j,j'}$ for the two spins are positive (negative) if $\Delta r = r_j - r_{j'}$ is parallel to the x direction. This implies that the spin moment direction for two nanoparticles should be aligned parallel along Δr . This picture works fine for 1D case, however, breaks down for a 2D square lattice situation when the particle at r_j has nearest neighbors at both x - and y - directions: How should the spin aligned? This may cause a spin frustration-like situation and lead to the paramagnetic ordering. Therefore, I doubt how they come up with the conclusion that a ferromagnetic ordering state. Is it just due to their assumption that only $\langle S_x \rangle$ is non-zero in their mean-field model?

In summary, I find the article is somewhat interesting, however, the physics is unclear to me and do not recommend the publication at this stage. If the authors can address my questions above, it could be considering for a publication.

I have read through the manuscript and the response letter. The authors had replied to most of the questions by the reviewers, however, I found some of the answers are not satisfactory, listed below:

1. In the questions Q-C1 and Q-C2 from the second reviewer, the reviewer had doubt about the effect on acoustic phonons. Although the authors claimed that the spin-phonon coupling is available to acoustic phonon by referring to Kittle and Chikazumi's work, such effect should only take place at non-zero wave vector. However, the current paper mainly consider the phonon at the Gamma point where the frequency of the acoustic phonons should be zero. So how does such "magneto-elastic" effect play the role for the acoustic phonon?

2. On Page 11, the third reviewer raised question on the validity of using $\langle S^2 \rangle = S^2$ in their mean-field approximation. I also have concern at this point. The authors' answer that they are "considering large spins, i.e. $S \gg 1$ " does not solve this question. They should somehow prove the validity of this approximation.

In my opinion, the authors had addressed most of the questions raised by the reviewers. I think the responses to the first and second reviewers are fine. However, I am not sure whether the third reviewer would accept the explanation, since his concerns, which are reasonable, seem to be quite critical. In this sense, I suggest the publication decision could be made if the third reviewer feels satisfied with their answers.

Reviewer #2:

Remarks to the Author:

The manuscript reports on a theoretical demonstration of magnetic order which relies on long-range spin-spin interaction which is in turn mediated by phonon-polaritons. The authors pin the emergence of this interaction to a magnetoelastic origin and argue, that for time reversal symmetric materials, such a long-range spin-spin interaction is one of two non-zero interaction contributions. In a setting in which the spin is sourced by nanoparticles in different arrangements, they explicitly find a first-order magnetic transition, as opposed to conventional magnets.

The manuscript shows the possibility of altering material properties by accounting for something as fundamental as vacuum fluctuations of the electromagnetic background, a research area which is timely and exciting. However the manuscript is far from nature communication standards for several reasons, as listed below.

First and foremost, the manuscript, which presents a highly approximated model Hamiltonian, is courageously overselling their findings, I will pick one example of that (which is already in the abstract!)

“Consequently, a large magnetization develops abruptly on lowering the 26 temperature which would enable magnetic memories admitting ultralowpower thermally-assisted writing [11, 12] while maintaining a high data.”

The story line is hard to follow because of several conceptual jumps that which are hard to grasp without reading the supplementary material.

Conceptually, from a broader perspective, I find three major pitfalls of the manuscript

There is no discussion of length scales: the interaction is said to be long-range but what is long-range in this case? How does the size of the nanoparticles play a role with respect to the long-wavelength approximation for example? And especially in the case in which the nanoparticles are put in a lattice, is the long-wavelength approximation still valid?

Clearly, as also pointed out by one of the previous referee, the interaction described in the manuscript has no effect for the spin $1/2$ particles. The reply of the authors to that is that one should then consider the case of $S \gg 1$. Now I wonder, for such a high spin one is most likely in the classical regime so does the exchange interaction make sense at all and would't it be a stretch to claim effects arising from exchange of virtual photons in this limit?

The integration of the photonic degrees of freedom that leads to an effective static contribution from the electromagnetic environment is not properly justified since the arbitrary cut-off used to perform the integral would not be that arbitrary anymore once the characteristic length scales of light and matter are considered. Furthermore it is not clear the role of free space fluctuations here.

Finally I agree with the most if not all of the objections of the previous reviewers, especially referee #2 and #3. In particular, as also pointed out by reviewer #2, there is a large body of literature discussing magnetoelastic coupling and there is no direct quantitative comparison in this manuscript of the strength of the claimed effect in relation to the other well investigated interactions. As already pointed out above also reviewer #3 has major doubts on the handling for the cut-offs used to derive the static approximation.

Therefore I do not consider the manuscript to be appropriate for publication in nature communications and it should be submitted to a much more specialized journal after thoroughly

clarifying the theoretical jumps made in the derivation of the model(s).

Reviewer #4 (Remarks to the Author):

We thank the reviewer for taking the time to assess our manuscript and for providing suggestions aimed at its further improvement.

The authors theoretically studied a so-called “spin-phonon-photon coupling” induced magnetic state in the nanoparticle system. Spin-phonon interaction couples the local spin of the nanoparticle to the vibrational mode, while an electromagnetic field collectively excites all the nanoparticles across the system, leading to an effective ferromagnetic coupling. The topic is interesting, however, I have a few questions upon reading the article.

First, the definition of “spin” in the paper is confusing to me. The spin operator S in the article is defined in the nanoparticle level, i.e. the net spin of all individual magnetic atoms within the nanoparticle. However, the “phonon” is defined in the atomic level when they derive the spin-phonon coupling Hamiltonian in the SI. Thus, the inconsistency between them causes confusion and the validation of H_{s-p} is questionable.

The reviewer has understood correctly that our spin operator corresponds to the net spin of the entire nanoparticle, while the phonons are considered at atomic level. This so-called macrospin approach is widely adopted in the treatment of magnetic nanoparticles. For example, see PRB 72, 014446 (2005) and references therein. It is useful and effective since the magnetization (or spatially resolved spin density) does not change with space on nanoscale distances and is the same throughout the particle. Thus, one can describe the magnetic properties in terms of the net spin, while starting with an atomic spin description.

As shown in the SI (for example, see Eq. S3 and subsequent equations) and is known from the previous literature on magnetoelastic coupling referenced in the SI, our approach starts with a microscopic and spatially dependent spin direction. It then exploits the spatial invariance of the spin in order to conveniently integrate the magnetoelastic coupling energy expressing it in terms of the total spin, or rather the direction of the spin. At the same time, we employ the atomic approach to the phonon for deriving the magnetoelastic coupling, similar to the previous literature cited in the Supplementary Note 1, while obtaining the desired parameters for the common strain-based description of the phonons.

In summary, H_{s-p} has been derived via the analysis detailed in the Supplementary Note 1. We have now clarified and resolved this confusion further in the same note.

On the other hand, if the “phonon” is about the vibration between neighboring nanoparticles, this is also problematic. Two issues are raised: a) Since their bonding are generally weak, the lattice translational symmetry across the system is easily broken, leading to the failure of a definition of “phonon”; b) Even if the nanoparticles are bound to a 2D periodic lattice array through some external field, e.g. a substrate, they cannot interact with the incident photon through the dipole interaction unless they are charged! In other words, this model does not work for charge-neutral particles. If the nanoparticles carry positive/negative charges, the Coulomb interaction term should be included in the model. Since the Coulomb interaction is generally larger than the long-range magnetic interaction by some orders of magnitude, I doubt whether the current conclusions still hold.

The phonon is indeed within a single nanoparticle. Hence, the above comment is not relevant.

Furthermore, in the lower panel of Fig.1, it shows that for a two-spin system, the xx (yy) component of $\Lambda_{j,j'}$ for the two spins are positive (negative) if $\Delta r=r_j-r_{j'}$ is parallel to the x direction. This implies

that the spin moment direction for two nanoparticles should be aligned parallel along Δr . This picture works fine for 1D case, however, breaks down for a 2D square lattice situation when the particle at r_j has nearest neighbors at both x- and y- directions: How should the spin be aligned? This may cause a spin frustration-like situation and lead to the paramagnetic ordering. Therefore, I doubt how they come up with the conclusion that a ferromagnetic ordering state is present. Is it just due to their assumption that only χ is non-zero in their mean-field model?

The system would be a paramagnet if the net mean field evaluated by summing over all the spins (in the considered 2D lattice) and components would come out to be zero. Our numerical summation finds the net mean field [see Eq. (6) and following text] to be nonzero and resulting in a finite ordering temperature below which the spins are ordered. This is indeed within the mean-field approach, which is adequate for investigating the presence of this phase transition.

It is common, for example in multisublattice magnets [see Physics Reports 229, 81 (1993), for example] or even simpler models of ferrimagnets, to have competing exchange interactions that may appear to cause frustration. Also here, the phase transition to magnetic order is determined by the net mean field that successfully accounts for these competing interactions.

In summary, I find the article is somewhat interesting, however, the physics is unclear to me and do not recommend the publication at this stage. If the authors can address my questions above, it could be considered for a publication.

I have read through the manuscript and the response letter. The authors had replied to most of the questions by the reviewers, however, I found some of the answers are not satisfactory, listed below:

1. In the questions Q-C1 and Q-C2 from the second reviewer, the reviewer had doubt about the effect on acoustic phonons. Although the authors claimed that the spin-phonon coupling is available to acoustic phonon by referring to Kittel and Chikazumi's work, such effect should only take place at non-zero wave vector. However, the current paper mainly considers the phonon at the Gamma point where the frequency of the acoustic phonons should be zero. So how does such "magneto-elastic" effect play a role for the acoustic phonon?

This has been carefully analyzed and detailed in the Supplementary Note 1. We do not relegate it to Kittel and Chikazumi, but only use their description of magnetoelastic coupling with acoustic phonons as a starting point.

Please note that we are focusing on an optical phonon, and not an acoustic phonon. The analysis leading up to Eq. (S22) and further supplemented by the subsequent symmetry arguments provides a derivation that goes beyond the previous literature. It demonstrates nonzero coupling between our considered optical phonon mode and the nanoparticle spin.

2. On Page 11, the third reviewer raised a question on the validity of using $S \gg 1$ in their mean-field approximation. I also have concern at this point. The authors' answer that they are "considering large spins, i.e. $S \gg 1$ " does not solve this question. They should somehow prove the validity of this approximation.

The third reviewer's concern, as he/she clearly stated, is made for the $S = 1/2$ case since the square of any spin component is unity in that case, due to its relation to the Pauli matrices. It is not valid for our case of $S > 1/2$.

It might help to see that in literature [e.g., PRB 102, 100402 (2020) and PRB 96, 22414 (2017)] it is standard to use terms like S_x^2 and so on to capture magnetic anisotropy. This is because of the

same reason that we have already stated: $S > \frac{1}{2}$. Since certain communities work mostly with $S = \frac{1}{2}$ and this can lead to confusion, we have further clarified this.

Most known ferromagnets have spin $S > \frac{1}{2}$ and therefore, the Weiss mean-field theory has been developed for arbitrary values of S and thus the square of the spin components does not become unity. The result is that one needs to use a Brillouin function that depends on the value of S and quickly converges to the Langevin function in the limit of large S . For our numerical demonstrations we have considered the self-consistent equation associated with the Langevin function corresponding to our considered case of $S \gg 1$. Nevertheless, our mean-field theoretical framework is general and it can be applied for any value of $S > \frac{1}{2}$.

A detailed discussion of this can be found in the book *Magnetization Oscillations and Waves* by Gurevich and Melkov (see Fig. 1.1 on page 6 and the corresponding discussion). It demonstrates the validity of mean field theory for any value of S and thus corroborates our mean-field treatment. To clarify this, we have added this reference to the manuscript.

Furthermore, we have never used $\langle S^2 \rangle = \langle S \rangle^2$. As can be seen in Supplementary Note 6, we employed the standard mean-field technique of expanding $S = \langle S \rangle + ds$, retaining terms up to the first order in ds , and then replacing ds by $S - \langle S \rangle$. We have further clarified this in the manuscript.

In my opinion, the authors had addressed most of the questions raised by the reviewers. I think the responses to the first and second reviewers are fine. However, I am not sure whether the third reviewer would accept the explanation, since his concerns, which are reasonable, seem to be quite critical. In this sense, I suggest the publication decision could be made if the third reviewer feels satisfied with their answers.

We believe that we have adequately addressed all the comments and hope for a swift publication of our manuscript.

Reviewer #5 (Remarks to the Author):

We thank the reviewer for taking the time to assess our manuscript and for providing suggestions aimed at its further improvement.

The manuscript reports on a theoretical demonstration of magnetic order which relies on long-range spin-spin interaction which is in turn mediated by phonon-polaritons. The authors pin the emergence of this interaction to a magnetoelastic origin and argue, that for time reversal symmetric materials, such a long-range spin-spin interaction is one of two non-zero interaction contributions. In a setting in which the spin is sourced by nanoparticles in different arrangements, they explicitly find a first-order magnetic transition, as opposed to conventional magnets.

The manuscript shows the possibility of altering material properties by accounting for something as fundamental as vacuum fluctuations of the electromagnetic background, a research area which is timely and exciting. However the manuscript is far from nature communication standards for several reasons, as listed below.

First and foremost, the manuscript, which presents a highly approximated model Hamiltonian, is courageously overselling their findings, I will pick one example of that (which is already in the abstract!)

Our model Hamiltonian is based on careful symmetry arguments and captures the essential physics in a broad range of systems. Most theoretical works start with a model Hamiltonian. In particular, we refer to a broad range of important works on magnetoelastic coupling which have employed such model Hamiltonians based on symmetries.

“Consequently, a large magnetization develops abruptly on lowering the 26 temperature which would enable magnetic memories admitting ultralowpower thermally-assisted writing [11, 12] while maintaining a high data.”

The above statement, and possibly further similar statements in the manuscript, aim to highlight the new avenues made feasible by our finding of a first-order phase transition. In our opinion, such outlook-oriented statements are useful for the readers and the scientific community.

In order to address the reviewer’s concern, we have rephrased the above quoted statement from “.. would enable ..” to “.. could enable ..”.

The story line is hard to follow because of several conceptual jumps that which are hard to grasp without reading the supplementary material.

Conceptually, from a broader perspective, I find three major pitfalls of the manuscript
There is no discussion of length scales: the interaction is said to be long-range but what is long-range in this case?

It is long range since it does not decay exponentially, but algebraically. In this sense, it does not have a length scale associated with it, which is commonly possible for an exponential decay.

We now clarify this in the revised manuscript.

How does the size of the nanoparticles play a role with respect to the long-wavelength approximation for example? And especially in the case in which the nanoparticles are put in a lattice, is the long-wavelength approximation still valid?

The long-wavelength approximation is valid for all frequencies for which the relevant optical wavelengths are sufficiently large compared to the extension of the nanoparticles. For higher frequencies, higher-order terms in the multipole expansion would have to be included, although the lowest-order electric dipole term still appears without any modification. However, for cases of interest in this work, the dominant contribution to the integral determining the spin-spin interaction comes from small enough frequencies that do not necessitate the higher-order terms of the multipole expansion. These relevant frequencies depend on the pairwise distance between the nanoparticles, but not on their overall geometrical arrangement in a lattice or similar.

In the revised manuscript, we have now added a Supplementary Note 7 explicitly showing that if the size of the nanoparticles is somewhat smaller than the distance separating them (i.e., the nanoparticles are not close to touching each other), the long-wavelength approximation is accurate.

Clearly, as also pointed out by one of the previous referee, the interaction described in the manuscript has no effect for the spin 1/2 particles. The reply of the authors to that is that one should then consider the case of $S \gg 1$. Now I wonder, for such a high spin one is most likely in the classical regime so does the exchange interaction make sense at all and wouldn't it be a stretch to claim effects arising from exchange of virtual photons in this limit?

The high value of spin limits the possibility of quantum fluctuations in the spin states. This means we cannot hope to find a quantum spin liquid in our system, although a classical spin liquid is allowed. This is consistent with our considering magnetically ordered classical states.

The spin magnitude bears no negative consequence on the role of photonic or polaritonic quantum fluctuations that mediate the spin-spin interaction.

The integration of the photonic degrees of freedom that leads to an effective static contribution from the electromagnetic environment is not properly justified since the arbitrary cut-off used to perform the integral would not be that arbitrary anymore once the characteristic length scales of light and matter are considered. Furthermore it is not clear the role of free space fluctuations here.

We do not use a cutoff in the integrals (and our results thus do not depend on the value of a non-existent cut-off). As discussed in our response to the question about the long-wavelength approximation above, we now explicitly show that the results are actually independent of such a cutoff (or in other words, converge quickly as a function of the cut-off frequency). Hence, there is no arbitrary choice involved in our analysis.

A confusion might have arisen from a misinterpretation of a comment by the reviewer 3, who suggested that an arbitrary frequency cutoff should be introduced to avoid divergences in the effective interaction of each phonon mode with itself. This is a different issue and discussion altogether.

For this issue, we already explained in our previous response that we instead use the standard procedure of subtracting the formally divergent free-space self-interaction (which, when treated more carefully only leads to a small energy shift analogous to the Lamb shift in atomic systems). After this subtraction, the resulting expressions do not diverge. We note as well that this subtraction is not necessary for the interaction between phonons in different nanoparticles, which does not diverge in any case, and which is the focus of the paper. It is thus not necessary to introduce an arbitrary cut-off, and we refrain here from doing so.

Finally I agree with the most if not all of the objections of the previous reviewers, especially referee #2 and #3. In particular, as also pointed out by reviewer #2, there is a large body of literature discussing magnetoelastic coupling and there is no direct quantitative comparison in this manuscript of the strength of the claimed effect in relation to the other well investigated interactions. As already pointed out above also reviewer #3 has major doubts on the handling for the cut-offs used to derive the static approximation.

We disagree with the above assertions. Reviewers 2 and 3 found our work suitable for Nature Physics, but suggested clarifications on a few technicalities. Reviewer 4 also finds our work interesting but requested further clarifications. We have addressed all the objections.

In the previous response, we already clarified the unique and novel nature of the phenomena that we consider in the present work. The reviewer 3 (not 2) indeed pointed out a few references that were also investigating some aspect of the magnetoelastic coupling. We already clarified, in the last response, how they did not compare to our work directly.

Therefore I do not consider the manuscript to be appropriate for publication in nature communications and it should be submitted to a much more specialized journal after thoroughly clarifying the theoretical jumps made in the derivation of the model(s).

As detailed in our response above, we have addressed all the concerns raised by all the reviewers and thus hope that our manuscript can be published swiftly in Nature Communications.

Reviewers' Comments:

Reviewer #1:

Remarks to the Author:

Dear editor,

The authors' response has solved most of my concerns and therefore, I suggest the publication of the paper in Nature Communications.